# Computer Tomography in the Diagnosis of Ovarian Cysts: The Role of Fluid Attenuation Values

**DOI:** 10.3390/healthcare8040398

**Published:** 2020-10-14

**Authors:** Roxana-Adelina Lupean, Paul-Andrei Ștefan, Mihaela Daniela Oancea, Andrei Mihai Măluțan, Andrei Lebovici, Marius Emil Pușcaș, Csaba Csutak, Carmen Mihaela Mihu

**Affiliations:** 1Histology, Morphological Sciences Department, “Iuliu Hațieganu” University of Medicine and Pharmacy; Louis Pasteur Street, Number 4, 400349 Cluj-Napoca, Romania; roxanalupean92@gmail.com (R.-A.L.); carmenmihu@umfcluj.ro (C.M.M.); 2Obstetrics and Gynecology Clinic “Dominic Stanca”, County Emergency Hospital; 21 Decembrie 1989 Boulevard, Number 55, 400094 Cluj-Napoca, Romania; amalutan@umfcluj.ro; 3Anatomy and Embryology, Morphological Sciences Department, “Iuliu Haţieganu” University of Medicine and Pharmacy; Victor Babeș Street, Number 8, 400012 Cluj-Napoca, Romania; 4Radiology and Imaging Department, County Emergency Hospital, Cluj-Napoca; Clinicilor Street, Number 5, 400006 Cluj-Napoca, Romania; andrei1079@yahoo.com (A.L.); csutakcsaba@yahoo.com (C.C.); 5Obstetrics and Gynecology Clinic II, Mother and Child Department, “Iuliu Hațieganu” University of Medicine and Pharmacy; 21 Decembrie 1989 Boulevard, Number 55, 400094 Cluj-Napoca, Romania; 6Radiology, Surgical Specialties Department, “Iuliu Haţieganu” University of Medicine and Pharmacy; Clinicilor Street, Number 3-5, 400006 Cluj-Napoca, Romania; 7Oncological Surgery and Gynaecologic Oncology, Surgery Department, “Iuliu Hatieganu” University of Medicine and Pharmacy, 400006 Cluj-Napoca, Romania; mariusemilpuscas@gmail.com; 8General Surgery Department, Institute of Oncology “Prof.Dr. Ion Chiricuta”, 400006 Cluj-Napoca, Romania

**Keywords:** computer tomography (CT), Hounsfield units, metastases, ovarian cyst

## Abstract

Pathological analysis of ovarian cysts shows specific fluid characteristics that cannot be standardly evaluated on computer tomography (CT) examinations. This study aimed to assess the ovarian cysts’ fluid attenuation values on the native (Np), arterial (Ap), and venous (Vp) contrast phases of seventy patients with ovarian cysts who underwent CT examinations and were retrospectively included in this study. Patients were divided according to their final diagnosis into the benign group (*n* = 32) and malignant group (*n* = 38; of which 27 were primary and 11 were secondary lesions). Two radiologists measured the fluid attenuation values on each contrast phase, and the average values were used to discriminate between benign and malignant groups and primary tumors and metastases via univariate, multivariate, multiple regression, and receiver operating characteristics analyses. The Ap densities (*p* = 0.0002) were independently associated with malignant cysts. Based on the densities measured on all three phases, neoplastic lesions could be diagnosed with 89.47% sensitivity and 62.5% specificity. The Np densities (*p* = 0.0005) were able to identify metastases with 90.91% sensitivity and 70.37% specificity, while the combined densities of all three phases diagnosed secondary lesions with 72.73% sensitivity and 92.59% specificity. The ovarian cysts’ fluid densities could function as an adjuvant criterion to the classic CT evaluation of ovarian cysts.

## 1. Introduction

Imaging has an important role in the assessment of ovarian cysts since a precise characterization of these lesions dictates their further management [1]. Probably the most crucial aspect of imaging in the evaluation of these lesions is differentiating between malignant and benign ones [2]. Ultrasonography (US) is the first imaging modality involved in investigating adnexal cysts, but in some cases, it cannot certify their benign or malignant nature [2,3], mostly due to its low specificity rates [4]. While magnetic resonance imaging (MRI) can bring additional information about the characteristics of an ovarian tumor [5], computer tomography (CT) is usually reserved for evaluation in pre-treatment planning of a suspected adnexal malignancy [6].

The CT features of ovarian cysts have been investigated in several studies in the literature. These features include these lesions’ behavior under contrast administration [7], and the imaging appearance of several histological entities [4,8]; even diagnostic workups have been developed [9]. Although correlations between morphological and imaging features of ovarian cystic lesions have also been researched [10], the radiological characteristics of the fluid content of these lesions remain relatively unexplored. Several imaging features (such as multilocularity, thickened cystic septations, internal vegetations, etc.) that are routinely evaluated on medical images are considered to advocate for malignancy [1]. But the presence of these morphological changes does not certify the diagnosis of malignancy, since many mixed-type benign ovarian lesions can mimic the features of a malignant mass [6]. Although ancillary findings (such as peritoneal implants, adenopathy, ascites, or organ invasion) can increase the diagnostic confidence of malignancy [4], these changes suggest an advanced stage of the disease. Moreover, the interpretation of medical images is subjective, being dependent on the examiner’s experience and training level. On the other hand, the pathological examination of ovarian cysts shows specific characteristics regarding the fluid composition (in types of cellularity or physical and chemical properties) that can be typical for a certain lesion or a histopathological group [11,12,13]. Thus, it is possible that these characteristics are reflected in CT images, and may carry additional diagnostic information, but it is extremely difficult to macroscopically assess.

These limitations created the need for means of a quantitative assessment of the information comprised in imaging examinations. On CT examinations, quantitative information can be extracted via the measurement of the attenuation values, expressed as Hounsfield units (HU). The attenuation coefficient is a measure of how easily a material can be penetrated by an incident x-ray beam, quantifying the amount of beam weakening by the material it is passing through [14].

The aim of this study was to characterize the fluid content of ovarian cysts by measuring its attenuation on the native (Np), arterial (Ap) and venous (Vp) phases of the CT examinations. The goal was to investigate the possibility of using these values as a non-invasive differentiation criterion between benign and malignant cysts, as well as between primary tumors and ovarian metastases.

## 2. Materials and Methods 

### 2.1. Patients

This Health Insurance Portability and Accountability Act-compliant, single-institution, retrospective pilot-study was approved by the institutional review board (ethics committee of the "Iuliu Hațieganu" University of Medicine and Pharmacy Cluj-Napoca; registration number, 50/11.03.19), and a waiver consent was obtained owing to its retrospective nature. From September 2016 to April 2019, a keyword search in our radiology database was conducted in order to identify all pelvic CT scan reports referring to ovarian cysts, by using the terms “ovary”, “ovarian” and “ovary/ovarian + cyst/cystic”.

The original search yielded 529 reports. Each report was then analyzed by one researcher, and all cases in which the report was not referring to a cystic lesion were excluded (*n* = 66). The remaining 463 studies were reviewed by one radiologist who confirmed the lesions, and the medical records of these patients were retrieved from the archive of our healthcare unit and investigated for disease-related data. 

No age limit was imposed since only adult patients are commonly referred to our health-care institution. Previously documented adnexal cysts, as well as CT examinations performed after an initial visualization of the lesions on US and MRI, were included. Only the earliest examination of each patient was considered, and subsequent CT scans of the same subject were excluded (*n* = 79). 

The other exclusion criteria were: solid ovarian lesions (*n* = 42), small cysts (a maximum diameter in the axial plane of less than 25 mm, *n* = 74), the presence of imaging artifacts within the lesions (*n* = 37), the lack of a final clinical or pathological diagnosis (*n* = 16), the lack of gynecological follow up (*n* = 83), and the lack of native, arterial and venous phases on the CT examinations (*n* = 62). Through this process, we were able to include ovarian lesions that were of cystic/mixed type, previously documented and followed by gynecologists, which measured at least 25 mm (to ensure an adequate placement of the regions of interest), as seen on CT studies, which included native, arterial and venous phases as well as a lack of artifacts. Information about the menopausal status and menstrual phase was inconsistent through patients’ medical records, and for this reason, this type of data was not included. The final study population comprised 70 subjects.

### 2.2. Reference Standard

From our institution’s electronic medical records, data about the patients’ pathological results, clinical notes, and surgical reports were retrieved. From our radiology information system, previous and subsequent imaging examinations and radiological reports were also investigated. 

There was no previously-set restriction about the type of entity (cystic follicles, corpus luteum, follicular cysts, corpus luteum cysts, or serous inclusion cysts) included in the physiological (simple) cysts’ group (SCs). Five SCs were developed on ovaries that were surgically removed and underwent pathological analysis along with the underlying disease. Nineteen SCs did not undergo pathological analysis but were included as they met one of the following criteria: resolution at the following imaging assessment, no more than 3 mm variations in their largest diameter for at least three months following the initial CT examination or the reduction in size at any following examination. Three simple cysts belonged to patients diagnosed with polycystic ovary syndrome and their regression was documented within 87.6 days (range 49–136 days) under medication. Six SCs underwent clinical and imaging follow-up along with another gynecological disorder (cervical dysplasia, *n* = 3; leiomyoma, *n* = 2; endometrial polyps, *n* = 1). The resolution of five SCs was noted at the following US examinations (mean time from CT to remission, 50.33 days; range 32–63 days). One cyst showed no variation in size at a CT examination performed at 92 days after the initial scan, and four cysts showed a decrease in size at the following US examinations (mean time from the CT scans to the US examination that showed a reduction in size, 45.4 days; range 36–58 days; mean size reduction, 4.12 mm; reduction range, 3.3–5.2 mm). None of the lesions from the metastases group underwent pathological analysis, but since their primary tumor was well documented, the imaging and clinical diagnosis were straightforward. All other ovarian cysts were surgically removed and underwent pathological analysis. The pathological examinations comprised macroscopic and microscopic assessments of the lesions. In some cases, the examination was supplemented with immunohistochemical analysis. One to three samples of solid tissues were collected, stained with hematoxylin and eosin, and analyzed microscopically. The liquid content of the cysts was not microscopically assessed in any of the cases included. However, some of the lesions benefited from a gross description of their content (Table 1). Patients were divided according to the final diagnosis of their lesions into benign cysts (*n* = 32), primary cystic tumors (*n* = 27), and ovarian cystic metastases (*n* = 11). To evaluate the difference between benign and neoplastic cysts, the patients of the latter two groups were combined forming the malignant cysts group (*n* = 38). 

### 2.3. CT protocol

All CT scans were performed on the same unit, Siemens Somatom Sensation, 16 slices (Siemens medical solutions, Forchheim, Germany). Before the scan, all patients were given an appropriate amount of water to drink to fill the bladder. The CT scan covered the region from the dome of the liver to the ischial tuberosity attachment. Contrast-enhanced CT images were obtained following the injection of 80–140 ml of non-ionic iodinated contrast material at a concentration of 350 mg/ml (iohexol (Omnipaque 350; Daiichi-Sankyo Health Care, Tokyo, Japan)) at a rate of 2 ml/s. The parameters of the CT scan were 120 kV, 200 mAs, slice thickness of 3 mm. Arterial and venous phase images were obtained during 25–30 s and 65–75 s delays, respectively.

### 2.4. Imaging Evaluation

All examinations were reviewed by one radiologist and one gynecologist (C.M.M. and A.M.M.), who were aware of the patients’ final pathological and clinical outcomes. When multiple ovarian cysts were observed within the same examination, the images were cross-referenced with the pathological and other imaging reports and medical data to confirm the selection of lesions that were previously documented. Only one lesion from each patient was selected and marked. If multiple lesions within each were previously documented, the researchers marked the one showing the largest diameter of the fluid component. All examinations were subsequently anonymized. 

On a dedicated workstation (General Electric, Advantage workstation, 4.7 edition), each examination was reviewed by two radiologists (C.C. and A.L., with at least 10 years experience in abdominal and pelvic CT), blinded to the clinical and pathological outcomes. The attenuation values for each lesion were obtained by placing a two-dimensional region of interest (ROI) on two different regions within the same cyst. Using synchronized slices, the ROIs were placed in approximately the same locations within each lesion on axial slices of native, arterial, and venous phases. Lesions’ walls, intra-cystic proliferations, septations, blood clots, or other solid components were avoided (Figure 1). The ROIs were placed to incorporate the largest portion of the fluid content, with no restrictions regarding their dimensions. Each of the radiologists conducted its own set of measurements on each phase, and the results were averaged for each lesion and subsequently used for statistical analysis. 

### 2.5. Statistical Analysis

The Mann–Whitney U test was conducted to compare the mean attenuation values recorded on the native (Nm), arterial (Am) and venous (Vm) phases between benign and malignant cysts and primary ovarian tumors and ovarian metastases. The same test was used to compare the values recorded by the same group (benign group, malignant group, ovarian metastases, primary tumors, serous carcinomas, and simple cysts) between the three contrast phases. A multiple regression using an “enter” input model was conducted to identify which Nm, Am or Vm are independent predictors of malignant cysts and metastases, respectively. The coefficient of determination (R-squared) was computed, and the diagnostic value of the prediction model was evaluated using receiver operating characteristics (ROC) analysis. The ROC analysis was also performed for the parameters showing statistically significant results (a ***p***-value of less than 0.005) at the univariate analysis that followed the comparison between the four groups. In addition, the area under the curve (AUC) was calculated with 95% confidence intervals (CIs).

The coefficient of variation from duplicate measurements was calculated to determine the reproducibility of the measurements conducted by the two radiologists, thus estimating the within-run imprecision. To analyze the distribution of samples and the measurement errors, the Passing–Bablok regression was conducted for every contrast phase. The scatter diagram and regression line and the residuals plot are shown. The unprocessed measurements recorded by the two radiologists are shown in Appendix A. 

## 3. Results

The median values measured by each group and histopathological entity are shown in Table 2. The univariate analysis (Mann–Whitney U test) results following the comparison of the attenuations values between different groups and contrast phases are displayed in Table 3. 

The multiple regression analysis showed that the measurements conducted on the Ap were independent predictors of malignant cysts, and the attenuation values measured from Np were independent predictors of ovarian metastases (Table 4). The highest AUC (0.756) in the differentiation of benign from malignant cysts was obtained by the densities measured on Ap. The prediction model could identify malignant cysts with higher sensitivity (89.47%) than the ones exhibited by the attenuation measured on each contrast phase (Table 5). When differentiating metastases from primary ovarian malignant lesions, the prediction model showed the highest sensitivity (92.59%) but also the lowest sensitivity (72.73%) compared to the ones exhibited by the attenuation measured on each contrast phase (Figure 2). The attenuation values’ dynamics depending on the contrast phase and lesion are shown in Figure 3 and Figure 4. 

The coefficient of variation of the measurements conducted by the two radiologists on the native phase was 29.69% (overall mean, 11.3; standard deviation, 3.35), 27.8% for the arterial phase (overall mean, 12.79; standard deviation, 3.55), and 36.01% for the venous phase (overall mean, 13.63%; standard deviation, 4.91). The Passing–Bablok regression results are shown in Table 6. The scatter diagrams and the residuals plots are displayed in Figure 5. 

## 4. Discussion

Our results show that the benign cysts group expressed higher attenuation values on the venous phase (Vm for the entire group, 9.01 HU; Nm, 7.95 HU; Am, 7.81 HU). The same pattern of attenuation dynamics was found in most entities included in this group (except for the serous cystadenofibromas). However, there was no statistically significant difference between the median densities recorded within this group (*p* = 0.45–0.8), as well as within SCs (*p* = 0.56–0.76) on all three contrast phases. Overall, the measurements of the entire benign group showed high attenuation values on the native and venous phases and low attenuation on the arterial phase. This model was encountered in the two serous tumors comprised in this group, but mucinous tumors and simple cysts showed higher values of both Vm and Am, compared to Nm. 

Both primary tumors and the entire malignant group showed the highest values on the arterial, followed by the venous and native phases. Primary ovarian cystic tumors held significantly higher Am (*p* = 0.03) and Vm (*p* = 0.04) than Nm. This pattern was followed only by clear cell carcinomas, while the rest of the primary tumors and metastases expressed higher density on the venous phase. Although the measurements conducted on all three contrast phases showed statistically significant results when comparing primary and secondary malignancies (*p* = 0.0005, 0.001, and 0.0008), the only independent predictor for metastases was the attenuations measured on the native phase (Table 4). However, the ROC analysis indicated that the Nm attenuation has a higher sensitivity (90.91%) but considerably lower specificity (70.37%) compared to the prediction model, which incorporated all three contrast phases (sensitivity, 72.73%; specificity, 92.59%) for the diagnosis of metastases. Overall, metastatic lesions showed higher median values on all phases compared to primary tumors. 

Both primary and secondary ovarian malignancies show common clinical and radiological features [15]. The differentiation between cystic ovarian metastases and primary tumors may be useful when an extra-adnexal malignancy is suspected, especially due to the different prognosis and management of the two types of injuries [16]. Although a contrast-enhanced CT (CECT) scan of the chest, abdomen, and pelvis is considered a standard evaluation in cancer of an unknown primary site [17], the only reliable method to distinguish between the two types of entities comes through the histopathological examination [18]. A study conducted by Douglas et al. [19] that investigated the imaging features’ capability of distinction between primary and secondary ovarian malignancies showed that CT was unable to differentiate them based on multilocularity, bilateralism or the presence of solid components. However, another study [20] suggested that CECT was able to distinguish between Krukenberg and primary tumors based on the intratumoral cystic components, especially if the cystic walls demonstrated strong contrast enhancement. Although these studies [15,16,17,18,19,20,21] also considered the morphological changes of the lesions, our results based only on the analysis of the liquid components are worth considering. The attenuation values may reflect more than the physical characteristics of the liquids (such as viscosity) because metastatic intestinal carcinomas have a macroscopic and microscopic appearance very similar to primary ovarian clear cell carcinomas, the distinction between the two being made only based on immunohistochemical analysis [21].

Our results show that benign and malignant cysts could be differentiated based on the average values recorded on all three contrast phases (Nm, *p* = 0.0083; Am, *p* = 0.0002; Vm, *p* = 0.0029), but the only measurements that functioned as independent predictors of malignant tumors were the ones conducted on the arterial phase. Based on Am values, malignant cysts could be identified with the same level of specificity (62.5%) but a lower sensitivity (86.84%) than the one expressed by the prediction model (sensitivity of the prediction model, 89.47%). Overall, the median values recorded on each contrast phase by the neoplastic lesions exceeded almost two times the one recorded by the benign lesions. Since the two entities do not share similar pathological appearances like the ones documented on primary tumors and metastases [21], it is less likely that the attenuation values indicate their histopathological appearance. This outcome could be justified by the biochemical characteristics, but also cellularity, as previously described as present in these types of fluids [11], which could have an increased impact on attenuation measurements. The attenuation values of the malignant cysts exceeded the benign ones, probably due to the higher contamination of the liquid contained by the premiums, a fact reported at both gross and microscopic examinations [12,13]. The pathological analysis of our included entities showed a predominantly serous content of the lesions comprised in the benign group, as opposed to the heterogeneous fluids seen in the malignant cysts group (Table 1). The latter aspect may be responsible for the high median attenuation values measured in malignant lesions. Previously published pathological reports indicate that serous carcinomas show cysts filled with retained secretions [22], and the fluid may appear turbid or bloody [11], usually containing papillary or glandular aggregates [12]. We identified the heterogeneous content of serous carcinomas, the dense content of mucinous cystadenocarcinoma, and the higher attenuation values recorded in metastases as the main reasons for the benign and malignant lesions’ successful differentiation.

The conventional CT evaluation of these cysts also encounters several pitfalls. This examination is not preferred in the characterization of adnexal masses, because of the hazards of ionizing radiation and the poor soft tissue discrimination [23]. However, the diagnostic power of this method for detecting ovarian malignancy is high. For US undetermined adnexal lesions, unenhanced CT can diagnose ovarian cancer with 76% (CI, 70–82%) sensitivity and 97% (CI, 95–98%) specificity. In the same scenario, the diagnostic rates increase with contrast administration, up to 81% (CI, 77–84%) sensitivity and 98% (CI, 97–99%) specificity [24]. Other reports show different sensitivity rates, such as the study conducted by Guo [25], which showed that CT had a sensitivity of 83.18% and a specificity of 85.25% in 168 patients. Altogether, our results, although showing medium-high diagnostic power for detecting malignant tumors, cannot outperform the classic CT evaluation of these lesions. Therefore, if further validated, our method can only function as complementary to the CECT assessment, especially when the morphological features advocating for malignancy are not present. 

The first report referring to the attenuation values expressed by ovarian cystic content was published in 1997 [26]. Although the exact details of the measurements’ workflow were not specified, the authors [26] noted the values corresponding to each histopathological entity that was evaluated. Thus, the reported mean attenuation values were: 12.4 ± 8 HU for simple cysts, 5.5 ± 5.2 for serous cystadenomas, and 61 ± 29.7 for serous cystadenocarcinomas [26]. We obtained lower values for simple cysts (Nm, 6.56 HU) and serous carcinomas (Nm, 9.34 HU), and higher values for serous cystadenomas (Nm, 23.95 HU). However, the authors [26] did not mention the phase that was used for measuring the attenuation values. Other sources of disagreement could include the different scanner (Siemens Somatom HiQS) or different acquisition protocol used; the latter consisted of administrating a different contrast agent (ioperamol (Iomeron 400, Bracco SpA, Milan, Italy)) in an initial bolus injection of 50 mL followed by a continuous pump infusion of 140 mL at a rate of 0.5 mL/s [26]. Our results are similar to the ones reported by another study [27] referring to the attenuation of the cystic portions of ovarian clear cell carcinoma, which showed an average value of 24.2 HU (range, 13–34 HU). In addition, a previous report [28] mentioned that simple cysts’ internal attenuation is generally less than 15 HU, while another [29] described benign ovarian cysts (including functional cysts) as having homogeneous internal densities near water, which mostly agrees with our results. 

The contents of an ovarian cyst may show high attenuation due to blood or debris accumulation, or collection from contrast enhancement [30]. In malignant tumors, bleeding can be caused by the cancer itself, abnormal tumor vasculature, tumor regression, or event anti-tumor treatments (chemo and radiation therapy), and other drugs commonly used in oncological patients (such as immunotherapies and anticoagulants) [31]. Two reports [30,32] noted that an ovarian cyst complicated by bleeding will show fluid-fluid or fluid-debris levels, with an attenuation range of 25–100 HU. It was previously documented that the CT appearance of the individual locules and within the same locule of an ovarian tumor may vary due to the differences in protein content and hemorrhage [15]. Most likely, these factors impacted the measurements conducted by the two radiologists, leading to several disagreements, as well as an unexpected attenuation drop between Nm and Am found within several of these lesions. Although the two researchers were instructed to target the same slice within the three contrast phases, it is possible that in many cases they chose different levels within the same cyst. Although only providing one slice for analysis would have led to closer results of the conducted measurements, this would not be a feasible approach for future clinical practice. On the other hand, because the included examinations were conducted on the same machine, we were able to counteract some factors that can influence the attenuation values such as aspect ratio, voxel size, and tube voltage [21].

The socio-economic aspect of adnexal cysts should not be neglected. The diagnosis of an ovarian mass causes anxiety in patients, which often pressurizes physicians to remove it out of fear that they have cancer [33]. This unnecessary surgery represents a significant cost to society, but also to patients because it increases the risk of ectopic pregnancies and may interfere with fertility [34]. On the other hand, CT examinations also do not come at low cost, and also expose the patients to radiation. But since our workflow consists of a simple ROI placement, this approach could be easily used as an adjuvant criterion for the conventional CT characterization of lesions already referred to this type of investigation, or when an adnexal cyst represents an incidental finding. It is possible that a combination of morphological assessment and fluid attenuation measurements would result in higher rates of correct diagnosis, but the evaluation of the premiums was not the goal of the current study, being already extensively investigated by previous research. 

Our study had several limitations. Firstly, due to its retrospective nature, it may have selection and verification bias regarding selected patients and gynecological follow-up, which mainly depended on the status of the institution and referral hospital. Secondly, the number of subjects included was relatively low, and the malignant group included about 20% more patients than the benign group. Another limitation was not including the patients’ menopausal status and the phase of the menstrual cycle. The latter was specified only in 4/24 medical records of the patients comprised in the SCs group. In addition, no direct correlation between the histological features of the cystic fluid content and the imaging appearance of these lesions could be performed since this analysis is not routinely performed in our healthcare center. Only a part of the lesions benefited from a gross description of their fluid content. However, a previous study [35] that investigated the imaging appearance of ovarian cysts stated that the choice of including only histologically proven lesions produced “greater selection bias”, decreasing the specificity of their reported findings. Therefore, the interpretation of the results from a histological point of view was also based on the literature data. We believe that for more accurate characterization of the fluid content, the attenuation values must be interpreted along with the cytological and biochemical features. The fact that two researchers (C.M.M. and A.M.M.) were aware of the final diagnosis can also be viewed as a limitation. However, this approach was necessary because at the time of the CT examinations, several patients had multiple lesions, and we wanted to include only the ones that were pathologically or at least clinically documented. After this stage, these investigators (C.M.M. and A.M.M.) did not intervene in any way in interpreting the images, reporting the results, or conducting the statistical analysis.

## 5. Conclusions

We demonstrated a statistically significant difference between benign and malignant lesions and primary tumors and metastases based on the attenuation measurements of the cystic component on native and contrast phases. Although successful, it is very probable that these results are a consequence of different biochemical and physical fluid properties, and not the lesions’ appurtenance to a certain histopathological group. This study opens the way for further research aiming to link the fluids’ attenuation dynamics to their inner components. If further validated, this approach can be useful to patients with incidental findings, as well as to the ones already referred to abdominopelvic CT for staging a suspected adnexal malignancy, as an adjuvant criterion to the classic CT evaluation of ovarian cysts.

## Figures and Tables

**Figure 1 healthcare-08-00398-f001:**
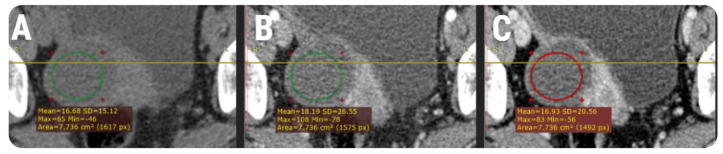
Schematic representation of the region of interest’s (green and red) placement on the native (**A**), arterial (**B**) and venous (**C**) phases using synchronized slices (yellow line) on a computer tomography examination of a 61-year-old patient with histologically proven clear cell ovarian carcinoma.

**Figure 2 healthcare-08-00398-f002:**
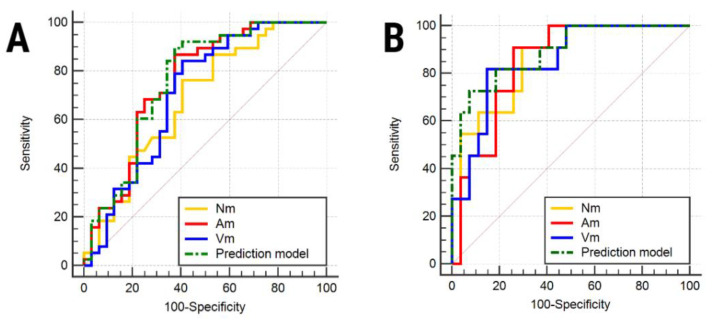
The receiver operating analysis curves following the mean attenuation values for the diagnosis of malignant cysts (**A**) and ovarian metastases (**B**). Nm/ Am/ Vm, mean values measured on the native/arterial/venous phases.

**Figure 3 healthcare-08-00398-f003:**
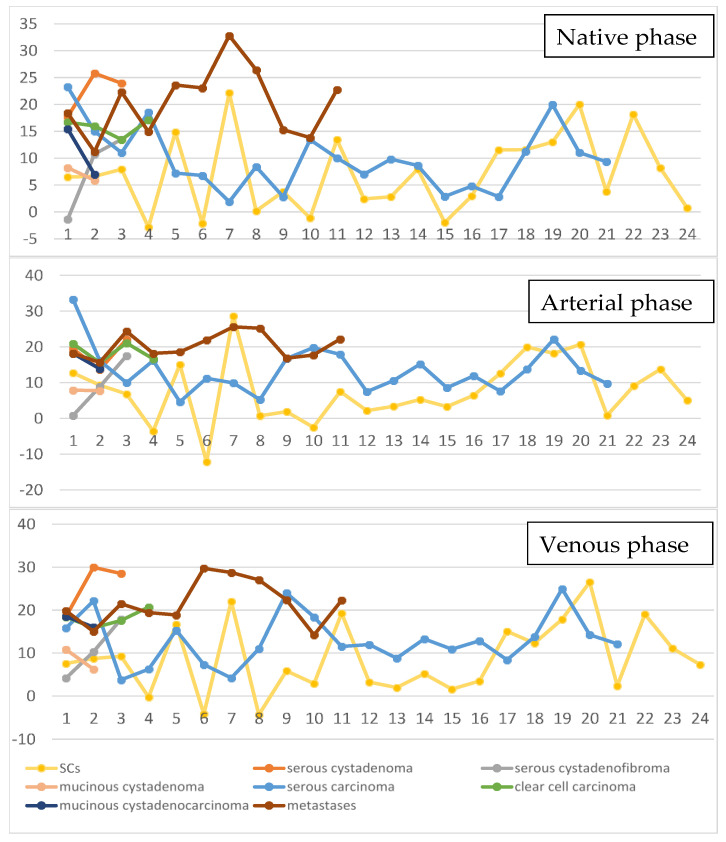
*Line and markers* diagram showing the averaged attenuation values measured for each histopathological entity on native, arterial phase and venous phases. The left vertical columns represent the attenuation values in the form of Hounsfield units. The horizontal lines represent the number of each case.

**Figure 4 healthcare-08-00398-f004:**
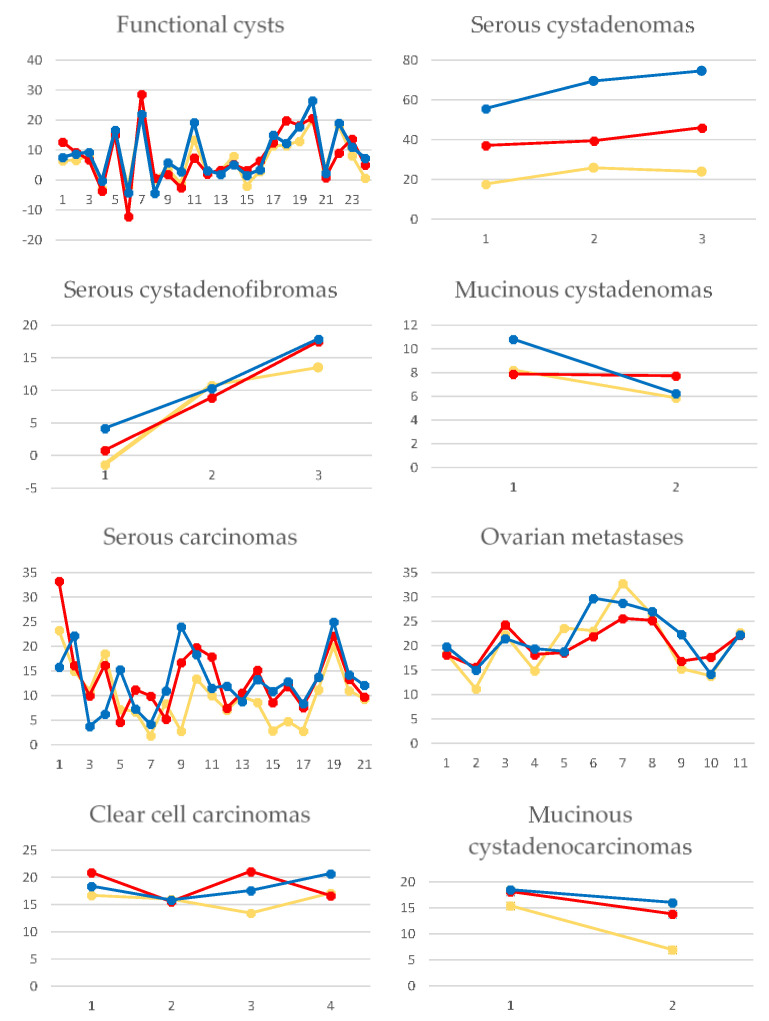
*Line and markers* diagram showing the averaged attenuation values measured for each case belonging to each histopathological entity on native (yellow lines), arterial (red lines) and venous (blue lines) phases of the computer tomography examination. The left columns represent the attenuation values in the form of Hounsfield units. The horizontal lines represent the number of cases.

**Figure 5 healthcare-08-00398-f005:**
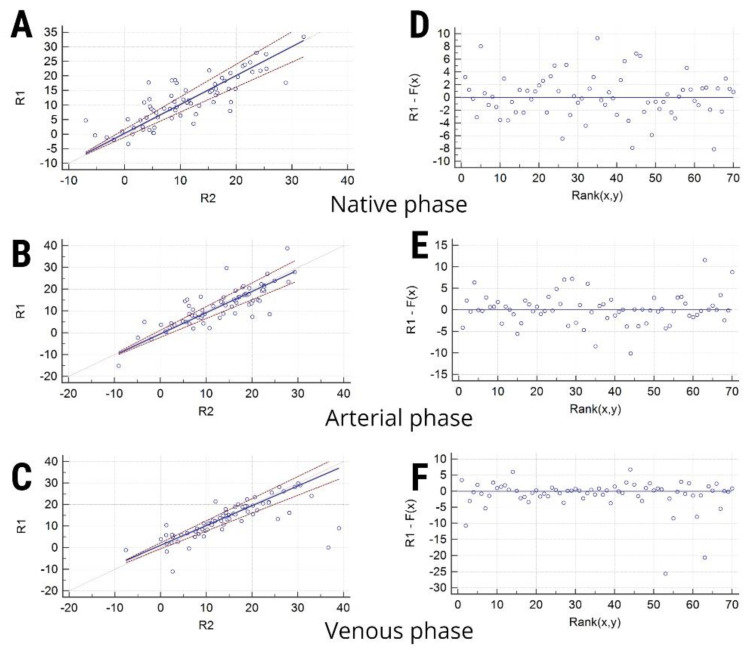
(**A**–**C**) Scatter diagrams that show the observations (solid line) and the confidence intervals (dashed lines) for the measurements conducted by the two researchers (R1 and R2) on each contrast phase. (**D**–**E**) Residuals plots allowing the visualization of the goodness of fit of the linear model by the measurements conducted by the two researchers on each contrast phase.

**Table 1 healthcare-08-00398-t001:** Patient groups, the time elapsed from the computer tomography (CT) examination until the pathological analysis, the main pathological findings from each group and the average diameter of every histopathological entity.

Group	Histological Entity	*n*	Pathologic Confirmation (d)	Cases with Available Description	Pathological Findings	Diameter (mm)
Benign cysts (*n* = 32)	simple cyst	24	43.2 ± 12.8	3	clear; slightly haemorrhagic; serous	31.28 ± 9.41
	serous cystadenoma	3	74.2 ± 21.4	3	serous; yellow	71.6 ± 26.76
	serous cystadenofibroma	3	38.2 ± 5.6	2	watery; serous	12.56 ± 8.25
	mucinous cystadenoma	2	18.5 ± 8.4	1	mucinous	135.05 ± 7
Primary tumors (*n* = 27)	serous carcinoma	21	29.5 ± 10.3	9	gelatinous; brownish; serous; slightly haemorrhagic	59.42 ± 24.07
	clear cell carcinoma	4	12 ± 4.8	3	clear; watery; turbid	54.06 ± 23.11
	mucinous cystadenocarcinoma	2	22.6 ± 5.2	1	thick; viscous	230 ± 36.3
Ovarian metastases (*n* = 11)	colon adenocarcinoma	1	-	0	-	40.3
	gastric adenocarcinoma	5	-	0	-	61.81 ± 48.76
	breast cancer	5	-	0	-	60.02 ± 12.72

*n*, number of patients; d, days.

**Table 2 healthcare-08-00398-t002:** The median attenuation values expressed by each group and histopathological entity. Values are expressed in Hounsfield units. Values in the brackets correspond to the interquartile range.

Group	Contrast phase		
	N_m_	A_m_	V_m_
benign cysts	7.95 (2.63–13.47)	7.81 (2.73–14.4)	9.01 (3.35–17.88)
primary tumors	10.04 (6.98–15.34)	13.84 (9.93–17.61)	13.8 (10.92–18.17)
metastases	22.31 (15.01–23.47)	18.58 (17.81–23.81)	21.49 (19.05–25.89)
malignant cysts	13.45 (8.4–18.4)	16.41 (11.21–19.7)	15.84 (11.96–20.7)
simple cysts	6.56 (1.58–12.28)	6.63 (2.03–13.24)	7.45 (2.62–15.87)
serous cystadenoma	23.95 (17.69–25.8)	19.44 (13.67–19.45)	28.52 (18.52–29.98)
serous cystadenofibroma	10.78 (−1.37–13.49)	8.87 (0.8–17.5)	10.31 (4.18–17.89)
mucinous cystadenoma	7.05 (5.88–8.22)	7.81 (7.73–7.89)	8.53 (6.25–10.82)
serous carcinoma	9.34 (5.78–12.34)	11.9 (4.6–16.45)	12.13 (8.6–15.65)
clear cell carcinoma	16.35 (14.72–16.91)	18.77 (16.10–21)	18 (16.73–19.55)
mucinous cystadenocarcinoma	11.21 (6.97–15.45)	16 (13.84–18.16)	17.27 (16.04–18.51)
colon adenocarcinoma	22.72	22.18	22.28
gastric adenocarcinoma	18.4 (13.06–22.95)	18.19 (16.91–21.47)	19.42 (15.02–21.49)
breast cancer	22.89 (13.81)	22.04 (17.69–25.21)	24.72 (22.28–28.76)

N_m_/A_m_/V_m_, median values measured on the native/arterial/venous phase.

**Table 3 healthcare-08-00398-t003:** The univariate analysis results, comparing the attenuation values of different groups **(A)** and of the same group between different contrast phases **(B)**. Values in bold are statistically significant. Np, native phase; Ap, arterial phase; Vp, venous phase.

**(A) Compared groups**	**Contrast phase**		
	Np	Ap	Vp
Benign and malignant	**0.0083**	**0.0002**	**0.0029**
Primary tumors and metastases	**0.0005**	**0.001**	**0.0008**
**(B) Group**	**Contrast phases’ comparison**		
	Np-Ap	Np-Vp	Ap-Vp
Benign	0.8016	0.4508	0.61
Malignant	0.0828	0.0856	0.8926
Metastases	0.947	0.7477	0.4385
Primary tumors	**0.0371**	**0.0476**	0.9655
Serous carcinomas	0.0575	0.0609	0.9298
Simple cysts	0.7728	0.5637	0.7650

**Table 4 healthcare-08-00398-t004:** Multiple regression analysis of attenuation measurements independently associated with the presence of malignant cysts and ovarian metastases. Bold values are statistically significant.

Predictors of Malignant Cysts						Predictors of Metastases			
Contrast phase	Coefficient	Standard error	*p*-value	VIF	Contrast phase	Coefficient	Standard error	*p*-value	VIF
N_p_	−0.01618	0.01492	0.2822	4.709	N_p_	0.03577	0.01470	**0.0204**	3.163
A_p_	0.03496	0.01286	**0.0084**	3.693	A_p_	−0.01525	0.01719	0.3813	3.070
V_p_	0.006727	0.01324	0.6131	3.928	V_p_	0.02151	0.01317	0.1116	1.997
Sign.lvl.	**0.0013**				Sign.lvl.	**0.0003**			
R^2^	0.2114				R^2^	0.4178			
R^2^ adjusted	0.1755				R^2^ adjusted	0.3664			
M.C. Coef.	0.8789				M.C. Coef.	0.6463			

Np, native phase; Ap, arterial phase; Vp, venous phase; VIF, variance inflation factor; R^2^, coefficient of determination; R^2^ adjusted, coefficient of determination adjusted for the number of independent variables in the regression model; Sign.lvl., significance level of the multivariate analysis; M. C. Coef., multiple correlation coefficient.

**Table 5 healthcare-08-00398-t005:** The receiver operating analysis’ results for the differentiation of benign from malignant cysts **(A)** and primary tumors from ovarian metastases **(B).** Values in the brackets correspond to the 95% confidence interval.

Compared Groups	Contrast Phase	AUC	Sign. lvl.	J	Cut-off	Sensitivity (%)	Specificity (%)
**(A)**	N_p_	0.684 (0.562–0.79)	0.005	0.36	>8.225	76.32 (59.8–88.6)	59.38 (40.6–76.3)
	A_p_	0.756 (0.638–0.851)	<0.0001	0.49	>9.3	86.84 (71.9–95.6)	62.5 (43.7–78.9)
	V_p_	0.708 (0.587–0.811)	0.0016	0.43	>10.82	84.21 (68.7–94.0)	59.38 (40.6–76.3)
	Prediction model	0.762 (0.646–0.856)	<0.0001	0.51	>0.4491	89.47 (75.2–97.1)	62.5 (43.7–78.9)
**(B)**	N_p_	0.865 (0.715–0.954)	<0.0001	0.61	>13.45	90.91 (58.7–99.8)	70.37 (49.8–86.2)
	A_p_	0.845 (0.691–0.942)	<0.0001	0.64	>16.725	90.91 (58.7–99.8)	74.07 (53.7–88.9)
	V_p_	0.852 (0.699–0.946)	<0.0001	0.67	>18.51	81.82 (48.2–97.7)	85.19 (66.3–95.8)
	Prediction model	0.892 (0.749–0.969)	<0.0001	0.65	>0.4678	72.73 (39–94)	92.59 (75.7–99.1)

Np, native phase; Ap, arterial phase; Vp, venous phase; AUC, area under the curve; Sign.lvl., significance level; J, Youden index.

**Table 6 healthcare-08-00398-t006:** Passing–Bablok regression results. R1/R2, researchers who conducted the measurements. RSD, residual standard deviation. Np, native phase; Ap, arterial phase; Vp, venous phase.

Contrast Phase	R1 (Mean ± SD)	R2 (Mean ± SD)	Systematic Differences (95% CI)	Proportional Differences (95% CI)	RSD (± 1.96 RSD Interval)	Test for Linearity
Np	11.06 ± 8.48	11.54 ± 8.16	0.35 (−0.95–1.69)	0.99 (0.85–1.17)	3.38 ± 6.64)	0.85
Ap	13.08 ± 8.47	13.26 ± 8.67	−0.56 (−2–1.15)	0.97 (0.86–1.08)	3.59 ± 7.04	0.97
Vp	14.37 ± 9.47	12.91 ± 8.27	1.13 (−0.55–2.24)	0.91 (0.82–1.02)	4.91 ± 9.63	0.97

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
