# Peer review of "Computer Tomography in the Diagnosis of Ovarian Cysts: The Role of Fluid Attenuation Values"

_healthcare, 2020, doi:10.3390/healthcare8040398_

Round 1
Reviewer 1 Report
General remarks
The Authors have examined the CT features of selected ovarian cyst’s fluid radiologic characteristics in 32 benign and 38 malignant tumors. These tumors were selected from 529 reports and 463 studies of cystic adnexal lesions. The main finding of this manuscript is that in the studied tumors there were statistically significant differences in native and contrast attenuation measurements of the cystic component between both groups. Ovarian cancer treatment results are much better when initial surgery is performed by a gynecologic oncologist, and the most likely reasons are complete surgical staging and optimal cytoreduction. Given these superior outcomes, new tests that could assist in the triage of adnexal masses and the appropriate referral for the treatment of patients affected with an adnexal mass are critical. Although the paper presents some new knowledge there are several important issues around this study that I wish to raise.
First, the definition of an “ovarian cyst” or “cystic lesion” is not presented in this manuscript. The complex lesions that the Authors describe were most probably cystic-solid lesions according to a currently used gynecological terminology. I do not understand why the Authors have introduced in their study an awkward and not widely used system of ovarian masses classification. The Authors do not convincingly explain why they have not used most popular texture features derived from the gray-level histogram, e.g. the tumor size, location, morphology, composition, number of cysts, growth pattern of the mural nodules, mural nodule HWR, enhancement of the mural nodules, ascites, CEA level, CA125 levels etc. for the cyst fluid content attenuation study. A current literature review indicates that based on these texture features, it is possible to distinguish between different types of adnexal lesions and also between healthy and various pathologically altered ovarian tissues with a very high degree of probability.
Second, it is well known that there are significant differences in recommendation patterns between gynecologists and radiologists. As compared with gynecologists, radiologists are typically more likely to recommend MRI or CT scans. The Authors state in their 1st sentence that “Imaging has an important role in the assessment of ovarian cysts….”. The truth is that not “any” imaging but pelvic and in selected cases abdominal sonography is the primary modality used to evaluate virtually all adnexal lesions. This difference can have important effects on resource use and patients' concerns. For instance, in 2009, the Society of Radiologists in Ultrasound (SRU) convened a consensus conference to discuss the management of adnexal cysts in asymptomatic women and reach consensus on when follow-up imaging was required. In a recent study, Bleau et al.[2018] have concluded that in clinical practice the use of the SRU guidelines on the management of adnexal cysts could dramatically decrease radiologist recommendations for follow-up of benign appearing cysts. More recently Levine at al. [Radiology 2019] updated the SRU statement and recommendations regarding reporting of simple adnexal cysts of suspected ovarian origin based on size and menopausal status. In the same year “A Consensus Guideline from the ACR Ovarian-Adnexal Reporting and Data System Committee” called “The O-RADS” has been presented by Andreotti et al. [Radiology, 2019]. The Authors do not seem to be aware of these important conclusions and recommendations of various radiological societies which have only recently admitted that due to its high costs and inevitable irradiation the CT scan will never be a first line examination in women with adnexal masses.
Third, the results of the study indicate that studying the cyst fluid attenuation has very little, if any new information in the context of predictive values. The sensitivity and specificity of this test even in the hands of experienced radiologists were relatively low and by no means could approach the currently used sonographic tests such like the IOTA group “Simple Rules Risks” or the “ADNEX” model, not mentioning the highest predictive values of pattern recognition that is used by highly experienced sonographers. When used for the comparison of the various predictive tests the Areas Under the ROC curves, i.e. AUCs for these methods always exceed the value of 0,90, in multiple external studies approaching even the values of 0,94-0,95. These values are far superior than the results in the presented manuscript where the highest obtained AUC=0,756. In the same time the “Simple Rules Method” has a very high positive LR (>20) and a very low negative LR (<0,08). Likelihood ratios (LR) should rather be presented along with the AUCs because they show how many times the odds increase of decrease when the test is positive or negative. There is no mention of these facts in the “Introduction” nor in the “Discussion” sections.
Fourth, there are multiple flaws in methodology with the strange composition of the studied groups. Not only multiple exclusions were made to the selected population but also strange criteria of cyst/lesions observation were used to confirm their benign character. The same problem appears with presumed metastases to the ovary that were only examined once during (possibly) primary surgery. Moreover, at least some of cancer metastases to the ovaries and/or female pelvis are solid and all such masses were excluded from the studied group. Some small tumors (<25mm in maximum size) were also excluded from the Author’s study, although it is not unusual to detect such small cancers at ultrasound examination. No detailed information on the construction of the Authors “prediction model” is presented. What were the important components of this model? The results are not clearly presented, there is no information why several specific small groups of tumors were compared apart from the Authors stated comparison of benign and malignant/metastatic ones. The diagrams presented in Figure 3 are misleading, there are no visible vertical lines as described below the figure, it is also not clear why the individual points presumably representing individual cases CT contrast attenuation values were connected with broken lines. The same misleading information is presented in Figure 4, where all cases are connected with broken lines for an unknown reason. The Authors present multiple statistical comparisons but they do not seem to understand what these numbers really mean. For instance, in Table 6 the R1 and R2 values are presented as means and SDs but the SD values are in all cases higher than 0,5 of the mean- in fact they approach 60-80% in some cases. Such results typically indicate a non-normal data set distribution and the results should be reported as medians (not SDs!) with interquartile range.
Fifth- there are multiple English language mistakes throughout the whole manuscript with mixing of the past and present tenses (e.g.lines 262, 265, 326) and using strange words outside their meaning (e.g. “premiums”, lines 299 and 361).
The Discussion section although very long, does not refer to the newer studies but deals with studies more than 15-20 years old. Interestingly, some of these studies (e.g. Kinkel et al.2005, [24], Guo et al. [25]) obtained higher predictive values of CT in the differentiation of adnexal masses than the data presented in the current Authors study. Wrong citation is used under the number [19], in line 275, where the Authors were not Douglas et al. but Brown DL et al. The conclusion section contains 4 sentences of which the first is not a conclusion but the repetition of the results. The second statement is only putative and contains the Author’s assumption, not proven by the studied data. The third sentence is far too optimistic because of multiple drawbacks and methodological flaws. The fourth statement should rather be that there is no further need to study retrospectively any cyst fluid CT contrast attenuation characteristics at least with the methods applied by the Authors.
The references used in this manuscript are in 78% of cases old and very old ones with only 8/35 published less than 5 years ago. Some references are old text books and internet publications like “Radiopaedia”, which should only exceptionally be used in scientific publications.
In summary, my main criticism is related to the serious methodological flaws that the Authors have made in their study. They have discarded the possibility that tumor morphology could add on top of cystic fluid content attenuation contrast CT imaging. There is a vast inconsistency both internationally, nationally and even between local institutions found in the description of ovarian masses. The use of non-standardized morphologic imaging descriptors and definitions often results in significant differences in subsequent interpretations. Since it is not clinically relevant to look only at separate fluid attenuation features this study adds very little, if any to the current state of knowledge. In case of ovarian cystic tumors analysis of prospectively acquired cyst fluid and CT scans should rather be performed. Furthermore, a predictive model combining clinical features with a cyst fluid CT features and perhaps malignancy and inflammatory markers could be applied to patient data.
Reviewer 2 Report
The aim of this paper by Lupean et al. was to characterize the fluid level of ovarian cysts by assessing their attenuation in the native (Np), arterial (Ap) and venous (Vp) phases of CT tests. Based on attenuation measurements of the cystic component at native and contrast stages, the paper highlighted a statistically significant difference between benign and malignant lesions and primary tumours and metastases. The manuscript provides substantial evidence for its conclusion and may serve as an interesting read to the journal’s readers. Major issues are missing information on conceptual advance over previously published work and scant discussion of results. It is important to revise the manuscript with the following comments:
Major comments
- The authors should extend the sections of introduction and discussion to show what additional knowledge this paper offers for advancing understanding and influencing the thinking in the field.
- It is important and useful to clearly indicate the inclusion criteria for the subject recruitment.
- The paper reported the observations recorded by two radiologists. For a robust conclusion, it would be important and useful to report the observations by three and more radiologists.
- The discussion segment would benefit from extending clinical significance, and referring the reader to potential adverse effects and drawbacks of this strategy.
Minor comments
- Page 3, line 106: Please fix the grammatical error in this sentence ‘..clinical notes, and surgical reports was retrieved..’
- Throughout the text, authors haven't consistent with the use of either American or British English. It would make better read with the use of consistent language wherever applicable. For examples, tumour (British), tumor (American).
Round 2
Reviewer 1 Report
The revised version of the manuscript still contains methodological drawbacks and raises serious concerns around the design of this research and the results presentation. First, the reasoning for performing such retrospective study remains obscure to me despite Author’s hopes, quote: “Also, please remember that our goal was not to impose a new diagnostic criterion for the evaluation of ovarian cysts, and we did not mention anywhere that the results of this study are of high clinical importance. Please consider that we aimed to evaluate an imaging aspect that was not studied before, and we transparently presented our results.” The main problem with such studies is that in fact they do try to introduce a new diagnostic criterion (see “Conclusions”), but the Authors when answering the Reviewers queries do not choose to clearly admit this fact.
Second, one of the main concerns is the use of multiple exclusion criteria which reduced the studied group from 463 to 70 cases i.e. only 15% of the whole group! The Authors try to explain this in a quite strange way, quote: “The strict exclusion criteria were needed, for example there was no use of including multiple examinations of the same patient, and also unconfirmed or solid lesions. The “strange criteria” for the confirmation of the benign nature of the included lesions are the ones commonly used in the practice of our healthcare unit.”. Although the Authors have provided detailed description of the reasons for multiple exclusions of their studied cases, they do not seem to understand that these exclusions may have introduced a heavy selection bias to this retrospective study. Another source of bias may be related to the lack of histological confirmation of some of the observed tumors. The medians of observation times indicate that it was typically 2-3 months and such time span may be too short to exclude malignancy or confirm benign tumor. Here again, the Authors have failed to answer the query why they have selected women who were not operated and why the tumors diagnosed as “benign” were not compared to “the gold standard”, i.e. histology. I would like to stress here again that for any reader familiar with gynecological pelvic imaging the term “cystic” refers to purely fluid filled lesions whereas solid parts require to change the name of the lesion to a “cystic-solid” mass. Obviously, if simple or smooth walled ovarian or paraovarian/paraoviductal cyst is detected at ultrasound examination, the CT imaging (with or without fluid attenuation study) is not necessary. Therefore, the whole problem presented in the manuscript i.e. distinguishing between the cystic fluid attenuation in benign and malignant types of ovarian cystic-solid masses in fact concerns a very limited and small group of women with difficult to characterize adnexal tumors. It seems that only a prospective study comparing various CT morphologic features with CT cystic fluid attenuation in only difficult complex adnexal cystic-solid masses could present some more clinically relevant results.
Third, the Authors present their adnexal cystic tumors classification method knowing that the predictive values are inferior to the existing non-invasive and less costly methods that do not use X-ray radiation. They claim that “It seemed only fair to compare the outcomes with the classic CT diagnostic performance in the diagnostic of ovarian malignancies, which we extensively address in the Discussion section…”. The belief of the Authors is apparently misleading because the results of CT fluid attenuation should be compared with “classical” CT features of benign and malignant tumors in the same group of women that were collected in this study. Moreover, they state that, quote: “Simultaneous integration of textural analysis, serum CEA levels, and lesion morphological features would have required a much more laborious analysis, which could hardly have been integrated into a single study.” Since the Authors obviously try to make “a scientific discovery” with their data it is surprising that although they have published recently a study “that investigated the texture analysis’ capability of differentiating between benign and malignant ovarian cysts”, there are no comparisons of the results in the presented/corrected manuscript. The “Discussion” section still lacks any information on the recent consensus statements between radiologists and gynecologists considering the ovarian cysts classification.
Fourth, the Authors seem not to be aware that there are important caveats to applying changes in cystic fluid attenuation on CT for the uterine adnexal tumor’s differentiation. Their answer is quote: “However, we acknowledge the CT limitations (lines 308-310 and 356-7). Moreover, we specified that the results of this study “can be useful to patients with incidental findings, as well as to the ones already referred to abdominopelvic CT for staging a suspected adnexal malignancy, as an adjuvant criterion to the classic CT evaluation of ovarian cysts” (lines 390-2)”. This seems to be quite strange and potentially dangerous strategy for the patients. The Authors further claim, quote: “In order to be able to do this extensively, together with a complete and complex statistical processing, we decided to focus on a single aspect of these lesions”. That is again a strange approach because at least some of the currently used tumor features should be compared in one study. The strange reasoning is presented in the Authors statement who state that, quote: “…our method does not aim to function as a per primam technique for evaluation ovarian cystic lesions. We simply proved that there is a statistically significant difference between the attenuation values expressed by the fluid component of ovarian lesions between different histopathological entities”. Although the presented findings may have only limited clinical values because of small numbers of cases (e.g. only 2 mucinous cancers) in specific tumors groups. Still, the Authors hope that their results are worth future studies, quote: “Regarding the predictive models, you can find in Table 5 that the diagnostic rates were not that low” whereas, in fact, they have proven the contrary.
Fifth, despite some description of the statistical methods used, the answers to the query asking how the logistic regression model was constructed and which variables were used in this model are still lacking. The Authors explanation to this issue is far too simplistic. Description of this “model” indicates that most likely all variables were "entered" and tested using multinomial logistic regression but without preceding univariate analysis of the statistical significance of each variable. Furthermore, the Authors claim that “The software used for the statistical analysis in this study (MedCalc) provides the data for this analysis in such a way. There is no method to change this information in median + IQR, perhaps only if removing this analysis from the study.” This really means that the Authors do not understand which statistical methods should be used for the specific data sets and the explanation of the software capabilities in unacceptable. Maybe they should consider the use of another software/statistician?
Finally, the Authors have refused to update any of their references although only 8 of 35 cited positions are less than 5 years old and some of them are still referring to the internet textbook-like sources (e.g.”Radiopaedia”). Most references are very old, typically published 15-25 years ago, which indicates their rather limited scientific value for the current medical research. The Authors explanation to this query is that they were unable to find newer publications on the studied topic is obviously not satisfactory. Maybe there are no publications because the manuscript topic was found not worth studying by other researchers so far?
Opinion summary. The manuscript is a retrospective analysis of the relatively small amount of cases studied with the relatively expensive and invasive imaging technique that is rarely used in clinical setting for the reason presented. Although the Authors responded in some way to most of the reviewer’s queries, these answers are unsatisfactory and the most important answers are still missing. The revised manuscript is still hampered by the serious methodological and theoretical flaws.
Reviewer 2 Report
The current revision and author’s response to previous comments are acceptable.